# Application of the Clustering Technique to Multiple Nutritional Factors Related to Inflammation and Disease Progression in Patients with Inflammatory Bowel Disease

**DOI:** 10.3390/nu14193960

**Published:** 2022-09-23

**Authors:** Agnieszka Dąbek-Drobny, Olga Kaczmarczyk, Agnieszka Piątek-Guziewicz, Michał Woźniakiewicz, Paweł Paśko, Justyna Dobrowolska-Iwanek, Aneta Woźniakiewicz, Aneta Targosz, Agata Ptak-Belowska, Paweł Zagrodzki, Małgorzata Zwolińska-Wcisło

**Affiliations:** 1Unit of Clinical Dietetics, Department of Gastroenterology and Hepatology, Jagiellonian University Medical College, 31-008 Krakow, Poland; 2Department of Gastroenterology and Hepatology, Jagiellonian University Medical College, 31-008 Krakow, Poland; 3Department of Analytical Chemistry, Faculty of Chemistry, Jagiellonian University, 31-008 Krakow, Poland; 4Department of Food Chemistry and Nutrition, Jagiellonian University Medical College, 31-008 Krakow, Poland; 5Department of Physiology, Faculty of Medicine, Jagiellonian University Medical College, 31-008 Krakow, Poland

**Keywords:** IBD, nutritional factors, SCFA

## Abstract

Diet and nutritional status affect intestinal inflammation in patients with inflammatory bowel disease (IBD). The aim of this study was to use a cluster analysis to assess structural similarity between different groups of parameters including short-chain fatty acid (SCFA) levels in stool as well as hematological and inflammatory parameters (such as serum C-reactive protein (CRP) and proinflammatory and anti-inflammatory cytokines). We also assessed similarity between IBD patients in terms of various biochemical features of disease activity and nutritional status. A total of 48 participants were enrolled, including 36 patients with IBD and 12 controls. We identified four main meaningful clusters of parameters. The first cluster included all SCFAs with strong mutual correlations. The second cluster contained red blood cell parameters and albumin levels. The third cluster included proinflammatory parameters such as tumor necrosis factor-α, CRP, platelets, and phosphoric, succinic, and lactic acids. The final cluster revealed an association between zonulin and interleukins IL-10, IL-17, and IL-22. Moreover, we observed an inverse correlation between IL-6 and body mass index. Our findings suggest a link between nutritional status, diet, and inflammatory parameters in patients with IBD, which contribute to a better adjustment of the nutritional treatment.

## 1. Introduction

A healthy diet, rich in various beneficial nutrients, affects human health through numerous mechanisms, including anti-inflammatory activity. The anti-inflammatory effects of a healthy diet were reported to reduce active inflammation in patients with inflammatory diseases [1]. Moreover, a proper diet helps maintain optimal body weight and avoid macronutrient and micronutrient deficiencies that can lead to health complications. Considering all these beneficial effects, a healthy diet is a crucial factor in the treatment of chronic intestinal disorders such as inflammatory bowel disease (IBD).

Inflammatory bowel disease is characterized by chronic inflammation of the gastrointestinal tract associated with the imbalance of the gut microbial community (so called dysbiosis). There are two main types of IBD: ulcerative colitis (UC) and Crohn’s disease (CD). Although the pathogenesis of IBD has not been fully understood, alterations in the gut microbiota and impaired immune response are known to be involved in disease onset and progression. Previous studies showed that active IBD is characterized by excessive levels of proinflammatory cytokines such as tumor necrosis factor-α (TNF-α) and interleukin IL-17 [2]. Furthermore, the microbiome imbalance seen in IBD leads to a disruption in the production of short-chain fatty acids (SCFAs) produced from dietary fiber [3,4]. This is essential because SCFAs play a key role in reducing the inflammatory process and maintaining intestinal homeostasis [5]. Recent reports showed that SCFAs regulate the immune response by releasing numerous cytokines, including TNF-α, IL-17, and IL-22 [6,7]. For example, it was demonstrated that SCFAs inhibited both IL-17 and IL-22 production by γδ T cells [6]. The proportion of IL-17 + γδ T cells and the quantity of synthesized IL-17 protein was reduced among γδ T cells in the presence of SCFAs, and the strongest effect was observed for propionate. Regarding to IL-22, a decreased production upon propionate exposure was found, especially among IL-17-producing γδ T cells [6]. They also strengthen the intestinal barrier and regulate the expression of tight junctions in epithelial cells [8]. Moreover, dietary factors (including the intake of dietary fiber and resistant starches) and the activity of intestinal inflammation affect gut microbiome composition and, consequently, SCFA production [5].

There is currently no single method of nutritional intervention in patients with IBD. Previous research described various enteral and parenteral nutrition strategies as well as numerous diets that could improve disease management. The role of gluten consumption in IBD remains unclear [9]. Studies have shown that gluten intake is associated with higher levels of zonulin, a protein that increases the permeability of the intestinal barrier [10]. Zonulin regulates intestinal permeability through the rearrangement of tight junction proteins [8]. However, the relationship between zonulin levels and gluten intake in IBD patients has yet to be described [9].

Diet helps maintain normal body weight, which is determined by body mass index (BMI). Malnutrition is a serious complication of IBD, which can also manifest as significant weight loss and reduced BMI [11]. These abnormalities are related to the severity of inflammation, the treatment applied, and disease-related changes in the daily diet. Poor nutritional status and low body weight also contribute to dysbiosis. Moreover, low BMI may be linked to more serious complications during treatment [12]. Owing to the natural course of IBD and the impact of drug treatment, particularly corticosteroids, patients are at risk both for weight loss and for rapid weight gain. Interestingly, the latest research showed that up to 40% of patients with IBD have obesity [13]. Until now, IBD has been associated mainly with low body weight (expressed as low BMI) and low body fat. At the same time, a new trend has emerged in the IBD population, revealing overweight and obesity in patients with malnutrition. So called malnutritive obesity, or “malnubesity”, is a new phenomenon that has been recently described in the context of numerous diseases. However, to our knowledge, data on malnubesity in IBD patients is lacking.

Considering the multiple nutritional and biochemical factors that affect the patient’s health status, and particularly the ongoing inflammation in IBD patients, we aimed to use the cluster analysis method to reveal important links between three groups of relevant parameters: (i) those related to nutritional status (such as BMI and albumin levels); (ii) SCFA profile; and (iii) proinflammatory and anti-inflammatory cytokines (IL-6, IL-10, IL-17, IL-22, and TNF-α).

## 2. Materials and Methods

### 2.1. Study Population

The diagnosis of IBD was established in line with the European guidelines based on clinical evaluation, endoscopic examination, and assessment of histological and radiological parameters [14]. Patients with UC had at least left-sided colitis, and those with CD presented with only colon involvement. Disease activity was assessed using the full Mayo criteria for UC and the Crohn’s Disease Activity Index for CD, based on which patients were classified into a group with active or inactive IBD. All participants underwent a clinical interview and completed a medical and nutritional questionnaire (a shorter version of the Food Frequency Questionnaire). Blood and stool samples were collected from all participants. In each patient, height and weight were also measured. To facilitate body weight classification, BMI was calculated. Participants were classified into three categories: (i) underweight (BMI < 18.5 kg/m^2^); (ii) normal weight (BMI, 18.5–24.99 kg/m^2^); and (iii) excessive weight (BMI > 24.99 kg/m^2^).

The control group included volunteers with functional bowel disorders who did not meet Rome IV diagnostic criteria for irritable bowel syndrome. The exclusion criteria were as follows: morbid obesity (BMI > 35 kg/m^2^), pregnancy, acute infections, any malignancies, alcohol and drug abuse, smoking, eating disorders, metabolic disorders, serious mental illness, celiac disease, gluten-free diet, prebiotic and probiotic intake, partial and total parenteral nutrition, and severe somatic disorders not related to IBD.

Written informed consent was obtained from all participants before the beginning of the study. The study protocol was approved by the Bioethics Committee at Jagiellonian University in Kraków, Poland (no. 1072.6120.18.2018; as of 23 February 2018). The study was performed in accordance with the Declaration of Helsinki.

### 2.2. Questionnaire

All study participants completed the medical and nutritional questionnaire. The first part included demographic and clinical data such as sex, age, disease duration, comorbidities, and medication use. Participants were asked about all drugs used in the previous three months, including over-the-counter medicines, vitamins, and dietary supplements, related both to IBD and to other medical conditions.

The second part of the questionnaire concerned diet. Participants were asked about their current diet over the previous three months. The nutritional questionnaire was a shorter version of the Food Frequency Questionnaire. In our study, the original version was limited to questions about the frequency of consumption of selected product groups: gluten products, fresh vegetables and fruits, dried fruits, whole-grain products, legumes, alcohol, and dairy products. Moreover, the frequency of fiber consumption was assessed according to participants’ responses. Participants were assigned to one of the two groups: low-fiber (<20 g/day) or normal-fiber (>20 g/day) consumption, depending on the type of diet used.

### 2.3. Identification of Fecal Organic Acids

The methods used for the identification and determination of fecal organic acids were described in detail in our previous study [15]. We collected stool samples and stored them at −80 °C for further analysis. The sample preparation and extraction processes were carried out at the Department of Food Chemistry and Nutrition, Faculty of Pharmacy, Jagiellonian University Medical College, Krakow, Poland. The preparation process included drying, milling, and the subsequent extraction process. All extracts were centrifuged and stored at −20 °C until further analysis. Capillary electrophoresis with spectrophotometric detection (CE-UV) was used for the determination of acids in stool samples. Electrophoretic measurements were carried out with the PA 800 CE apparatus plus the Pharmaceutical Analysis System (Beckman-Coulter, Brea, CA, USA) equipped with an ultraviolet spectrophotometric detector, at the Laboratory for Forensic Chemistry, Faculty of Chemistry, Jagiellonian University. The separation of the analyzed compounds was carried out in a fused silica capillary at 25 °C with −30 kV applied. The injection was made hydrodynamically by applying a pressure of 3.45 kPa for 8 s. The indirect spectrophotometric detection was performed at 230 nm. The separation buffer was composed of 1% of methyl-β-cyclodextrin in a commercially available buffer (Anion Kit 5, Analis, Namur, Belgium). The Anion Kit 5 was used for the qualitative and quantitative analysis of anions of inorganic and organic acids. The modification of the commercial separation buffer with cyclodextrin additionally allowed the separation of butyric acid anions from isobutyrate and valerate from isovaleric acid anions.

### 2.4. Identification of Inflammatory Parameters and Cytokines in Serum Samples

Biochemical tests, including the measurement of complete blood count, C-reactive protein (CRP), albumin, and calprotectin levels, were performed at the Department of Diagnostics of the University Hospital in Krakow, Poland, in accordance with relevant laboratory procedures.

Serum cytokine (IL-6, IL-10, IL-17, IL-22, and TNF-α) and zonulin levels were determined at the Department of Physiology, Faculty of Medicine of Jagiellonian University Medical College, using commercially available enzyme-linked immunosorbent assay kits according to the manufacturer’s protocol.

### 2.5. Statistical Analysis

#### 2.5.1. Comparisons between Groups

The differences between groups were assessed using the Kruskal–Wallis test with a post-hoc Dunn test. A probability level of less than 0.05 was considered significant. For the comparison of two groups, the Mann-Whitney test was applied. The Pearson chi-square test was used to check whether the observed difference in the incidence of active disease between UC and CD patients occurred by chance.

#### 2.5.2. Cluster Analysis

A hierarchical agglomerative cluster analysis was applied to parameters and subjectss (patients). The main aim of cluster analysis in our work was to place studied subjects (both—parameters and patients) into not a priori defined groups (clusters) in such a way that subjectsin a given cluster are more similar (close) to each other (for example, the parameters change in a similar way and patients have a similar parameter profile, respectively), and, contrary to that, the subjects in different clusters tend to be dissimilar (in the same sense). Although there are different mathematical measures (representations) of similarity (dissimilarity), the reasonable assumption is that the parameters more tightly clustered are more likely to represent similar patterns. In the hierarchical agglomerative method of cluster analysis, clusters are gradually extended by adding further individual subjects or previously formed clusters, where there is a rule that one cluster may be entirely contained within another one at each level of hierarchy, but no other kind of cluster overlapping is allowed.

The analysis was performed after the standardization (z-transformation) of all continuous data to obtain a zero mean and a unit variance for each parameter. Thus, for every data value, the mean of the respective parameter was subtracted and the result was divided by the standard deviation of the same parameter. Normalization was conducted to reduce the effects of spreads in individual parameters and to equalize the contribution of all parameters to the Euclidean distance used as a function of the distance in a clustering procedure. The Ward’s agglomeration method was used for grouping parameters, and average linkage was used for grouping patients. Ward’s method aims to minimize the sum of squares deviations inside clusters. At each stage of agglomeration, out of all possible pairs of clusters, the one is selected that gives a cluster with minimal differentiation, as is assessed by the analysis of variance method. In both attempts, the Mojena’s rate was calculated as a criterion to select the optimal grouping result.

#### 2.5.3. Correlation Analysis

The Pearson or Spearman correlation coefficients were calculated for the pairs of parameters, as appropriate. A probability level of less than 0.05 was considered significant. Statistical analyses were carried out using Graph Pad Prism v.3.02 (GraphPad Software, San Diego, CA, USA) and STATISTICA v. 13.3. (TIBCO Software Inc., Palo Alto, CA, USA). The STATISTICA package was also used for the graphic representation of data. The software available at http://statpages.org/ctab2x2.html (accessed on 18 July 2022) was used to perform the Pearson chi-square test.

## 3. Results

### 3.1. Descriptive Statistics

A total of 48 participants were enrolled in the study, including 26 patients with UC (mean age, 37 ± 4.5 years), 10 patients with CD (mean age, 29 ± 7.9 years), and 12 controls (mean aged 30 ± 15.2 years). The demographic and clinical characteristics of participants are shown in Table 1.

Participants were divided into three groups based on the BMI category: 12.5% of patients were underweight, 66.7% had normal weight, and 20.8% were overweight. In the UC group, 11.5% of patients were underweight and 27% were overweight, while in the CD group, 30% of patients were underweight and 20% were overweight. The remaining patients in each group had normal body weight. In the study population, 23% of patients followed a low-fiber diet, including 11.5% in the UC group and 40% in the CD group.

Nine acids were identified in fecal samples: succinic, acetic, lactic, propionic, butyric, isobutyric, valeric, isovaleric, and phosphoric. The results are presented in Table 2. Serum cytokine (IL-6, IL-10, IL-17, IL-22 and TNF-α) and zonulin levels are demonstrated in Table 3.

### 3.2. Correlation Analysis

Dietary fiber and gluten intake were not correlated with zonulin levels (data not shown). Similarly, there was no correlation between the type of diet and zonulin levels. BMI was inversely correlated with IL-6 (R = −0.34, *p* = 0.017 for the whole group and for participants with IL-6 above the lower limit of detection [LOD]: R = −0.684, *p* = 0.029, n = 10).

Interleukins IL-10, IL-17, and IL-22 levels were positively correlated with zonulin levels (R = 0.93, R = 0.80, and R = 0.88, respectively; *p* = 0.000 for all correlations). In addition, IL-17 was positively correlated with phosphoric acid (R = 0.30, *p* = 0.038). At the same time, no other correlations were found between fecal acid levels and cytokine and zonulin levels. There were no differences in zonulin levels between patients with IBD and controls (also separately for UC and CD groups).

### 3.3. Cluster Analysis

A dendrogram depicting the hierarchical clustering of the studied parameters is shown in Figure 1. Four main meaningful clusters were revealed. Cluster A included only SCFAs: isobutyric, isovaleric, propionic, butyric, valeric, and acetic acids. Biochemical parameters (albumins, hematocrit, hemoglobin, and red blood cells) were included in cluster B. Cluster C consisted of cytokines (IL-22, IL-10, and IL-17) and zonulin. Cluster D contained the most diverse parameters. It showed a larger dispersion of elements in the hypersphere than the other clusters and was split into two subclusters. Subcluster D1 contained BMI, TNF-α, and succinic acid, while subcluster D2 contained lactic acid, CRP, platelets, phosphoric acid, and white blood cells. These variables reflect the very different aspects of human physiology.

The parameters in each cluster were highly correlated with each other. For cluster A, the correlation coefficients ranged from 0.510 (butyric vs. isobutyric acid) to 0.930 (propionic vs. isovaleric acid), with a mean value of correlation coefficients equal to 0.775 ± 0.133. For cluster B, the correlation coefficients ranged from 0.569 (albumins vs. red blood cells) to 0.957 (hemoglobin vs. hematocrit), with a mean value of correlation coefficients equal to 0.775 ± 0.133. For cluster C, the correlation coefficients ranged from 0.796 (IL-17 vs. zonulin) to 0.945 (IL-10 vs. IL-22), with a mean value of correlation coefficients equal to 0.901 ± 0.052. In cluster D, the correlation structure was more varied. No correlations were shown for TNF-α and succinic acid with any other parameter, while BMI correlated only (inversely) with CRP. A positive correlation was shown for CRP with three mutually correlated parameters: platelets, lactic acid, and phosphoric acid. White blood cells correlated only with phosphoric acid. In the whole cluster D, significant correlation coefficients ranged from −0.373 (BMI vs. CRP) to 0.591 (CRP vs. platelets), with a median value of correlation coefficients equal to 0.327.

A dendrogram of similarity among participants is shown in Figure 2. The distribution of patients among clusters did not reveal any larger homogeneous clusters that would closely reflect the classification of patients into clinically different groups, regarded as disjoint information categories (Figure 2). In other words, the dissimilarities between patients labelled with different classification codes and characterized by a selected set of parameters were too small to generate non-fuzzy clusters corresponding strictly to the circumstances of the study.

However, a careful examination of the plot revealed that the two-elemental clusters appeared only for patients with active UC (code 2) and for controls (code 5). This means that the appropriate counterparts who “resemble” one another were provided via cluster agglomeration only for these two groups. A closer look at the graph provided by the cluster analysis also revealed patients with the most distinctive features compared with the rest of the population. They formed three one-element clusters, numbers 1, 2, and 4. Among the whole group of patients, in the unique cluster 1, the patient with the most distinctive characteristics had the highest values of the following parameters: acetic, phosphoric, butyric, valeric, and isovaleric acids as well as LOD for phosphoric acid. The patient from the next remote cluster 2 was distinguished by the highest levels of IL-10, IL-17, IL-22, and zonulin, the second highest results for succinic acid, and the third highest results for TNF-α and isovaleric acid compared with the whole group. Similar to the patient from cluster 1, they also had the highest LOD for phosphoric acid. The patient from cluster 4 was characterized by the highest value for isobutyric acid, the second highest values for isovaleric and propionic acids, and the fourth highest value for butyric acid. The five patients from cluster 3 belonged to active UC and active CD groups (codes 2 and 4, respectively). They had two- to three-fold higher mean values of TNF-α, IL-10, IL-17, IL-22, zonulin, and lactic acid compared with the remaining patients. At the same time, they had much lower mean levels of butyric (15-fold), isobutyric, and isovaleric acids (about five-fold).

## 4. Discussion

The use of cluster analysis allowed us to identify associations between nutritional status, SCFA profile, as well as inflammatory parameters, including serum cytokine levels. From an extensive body of clinical and biochemical data on IBD patients, we identified four groups of parameters in such a way that the parameters included in the same cluster were more similar to each other (that is, they changed in a similar way) than to those in other clusters.

We noted a link between propionic, butyric, valeric, acetic, isobutyric, and isovaleric acids. All of them are SCFAs or branched chain fatty acids (isobutyric and isovaleric acid) and have anti-inflammatory properties. The beneficial effects of SCFAs, especially butyrate, are due to their role in regulating regulatory T cells and strengthening the intestinal barrier by regulating tight junctions [16]. Previous studies showed that patients with IBD have reduced amounts of SCFA-producing bacteria, such as *Roseburia* spp. and *Faecalibacterium prausnitzii*, as well as reduced SCFA levels [17]. Similarly, in our previous study, we showed that patients with IBD (especially those with active disease) had low SCFA levels, including acetic, butyric, and valeric acids [15]. It seems that increasing the concentrations of SCFAs and the amount of dietary fiber, which is both a substrate to produce SCFAs and a modifier of the gut microbiome, may be an important target in IBD therapy [18].

In addition, we noted that red blood cell parameters were positively correlated with albumin levels. Anemia is the most common extraintestinal complication of IBD, affecting about one-third of patients [19]. The common causes of anemia include anemia of chronic disease, which results in decreased erythropoiesis and impaired iron metabolism [20]. This group of patients appears to have a higher risk of malnutrition, and reduced albumin levels are one of the indicators of it. It seems that patients with anemia, even without other clinical symptoms of IBD activity, should receive dietary counseling to counteract progressive malnutrition. In addition, hypoalbuminemia, along with anemia, may reflect severe intestinal inflammation, which is also a risk factor for malnutrition [14].

Importantly, we also found a positive correlation between zonulin and IL-17 levels, which is consistent with earlier studies [21]. Increased zonulin levels are associated with inflammation and disruption of the intestinal barrier (“leaky gut”). This disruption of the colonic mucosal cell lining was shown to induce differentiation of TH17 cells, releasing the proinflammatory cytokine IL-17 [22]. However, we did not show a relationship between dietary gluten intake and zonulin levels. It may be that increased zonulin release is related to dysbiosis in IBD patients, rather than gluten consumption itself, but this requires further research. It should be also noted that the enrolled patients showed lesions only in the large intestine. Further research should evaluate patients with CD involving the small intestine. There may also be other dietary and lifestyle factors that affect zonulin levels, for example, a Western diet and a low variety in the regular diet. The dietary intake of anti-inflammatory substances such as polyphenols may have a positive effect on the intestinal barrier and thus reduce zonulin levels. These food products include grapes, blueberries, olive, nuts, cocoa, fresh fruits, and legumes, but they are not commonly consumed by patients with IBD [23]. Additionally, butyric acid, the production of which is related to dietary fiber consumption, decreases zonulin secretion and thus reduces intestinal permeability [10]. However, in our study, we found no correlation between butyric acid and zonulin levels (data not shown). In other studies, 30% of IBD patients had non-celiac gluten sensitivity, and many of them followed a gluten-free diet [9]. Gluten is known as a protein that may promote intestinal inflammation and increase intestinal permeability; however, there have been no prospective studies evaluating the role of a gluten-free diet in the induction and maintenance of CD and UC [9].

We also observed correlations between proinflammatory factors: TNF-α, phosphoric, succinic, and lactic acid, CRP, platelets, and white blood cells. Interestingly, this group also included BMI, which was negatively correlated with CRP. High CRP levels, leukocytosis, and thrombocytosis are well-known markers of active IBD [14]. It seems that low BMI is a late marker of chronic severe intestinal inflammation in IBD. This may be particularly useful in patients in whom disease flare is not associated with a significant increase in the levels of other inflammatory parameters such as CRP or white blood cells, but who require the intensification of treatment of the underlying disease together with nutritional therapy. Lactic acid is another well-known metabolite that shows an increase in active intestinal inflammation. Succinic acid also appears to be a proinflammatory mediator, but the exact pathomechanism of its action is still unknown. Recent research showed that high levels of succinate may be associated with chronic inflammation in IBD [24]. Similarly, in our earlier study, we noted a trend towards higher levels of succinic acid in IBD patients [15]. Moreover, our observations are in line with previous research showing that the intake of inorganic phosphate can exacerbate inflammation. We found a positive correlation between phosphoric acid and well-studied markers of inflammation in IBD, such as CRP and leukocytosis. Inorganic phosphate is found in a diet rich in processed foods and additives. It can lead to an increase in the levels of proinflammatory cytokines through the activation of nuclear factor κB in the inflamed colon [25,26]. We also found a positive correlation between phosphoric acid and proinflammatory IL-17, which is associated with the nuclear factor kB pathway. However, the mechanism underlying the effects of phosphoric acid on intestinal inflammation is not fully understood. It appears that the determination of fecal organic acid concentrations, including lactic, succinic, and phosphoric acids, may serve as an important noninvasive marker of disease activity [27].

In the context of similarities between participants, we distinguished patients with active inflammation in the course of IBD who were characterized both by higher levels of TNF-α, zonulin, and lactic acid and by lower levels of butyric and isovaleric acid. Interestingly, apart from showing moderate or severe disease activity, they were all underweight. This finding is consistent with our hypothesis that the organic acid profile and evaluation of body weight are useful in assessing disease activity. Of note, the above similar parameters did not include biochemical tests such as CRP or leukocyte levels. This confirms previous results that the classic inflammatory markers are not useful for assessing disease activity in some IBD patients [28].

We observed an inverse correlation between IL-6 and BMI, which can be understood in two ways. On the one hand, low body weight and the associated malnutrition can exacerbate inflammation, expressed by elevated IL-6 levels. Our previous research showed that underweight patients were more often treated with steroids and antibiotics, which would suggest a greater disease severity in this group [22]. On the other hand, severe inflammation can contribute to problems with maintaining normal body weight and proper nutritional status. It is commonly known that high levels of proinflammatory mediators are cachectogenic and anorexic [28]. A routine screening of IBD patients for malnutrition is recommended, but current malnutrition screening tools do not incorporate IBD-specific characteristics and may be less adequate in this population [29] A correlation between BMI and inflammation was also described by Fiorindi et al. [30]. Interestingly, only 32% of respondents with malnutrition had a low BMI [30]. The remaining participants had a BMI within a reference range, which reflects the phenomenon of masking poor nutritional status by normal body weight or even excessive body mass [30]. Similarly, in our study, as many as 28% of IBD patients were overweight and obese, while only 10% were underweight. Therefore, considering the relationship between BMI and proinflammatory IL-6, it seems important to assess IBD patients for body weight using a body composition analyzer and not only weight measurement. According to recent research and our observations, too low a body weight is associated with a poorer prognosis of IBD and increases the risk of disease complications [31].

Our cluster analysis for patients was not able to distinguish specific clusters that would clearly differentiate patients. However, we observed some clustering trends for patients in terms of the type of disease (UC or CD) and disease activity. It seems that patients with CD and UC, despite numerous common features, including disease site, should be considered separately in the context of the influence of nutrients and nutritional status on inflammation. This may be due to differences in microbiota disturbances. On the other hand, the results of cluster-1 patients with distinctive characteristics seems interesting. It was a healthy person, with normal fiber consumption and a proper BMI, that was not taking any medications. The patient from the second cluster was a 50-year-old male with nonactive UC, overweight, and on a low-fiber diet. Cluster 3 included five patients with an exacerbation of IBD and low body weight at the same time (BMI, 18–19 kg/m^2^). It seems that the coexistence of similar nutritional factors and inflammation may be a factor classifying patients with IBD into certain subgroups. However, in the context of other discrepancies between participants, an individualized approach to patients seems important. The small number of participants and the heterogeneity of the study population are the main limitations of our study. Nevertheless, our study represents preliminary research that will form the basis for future studies on this topic.

## 5. Conclusions

This study in patients with IBD suggests a link between nutritional status, functional fiber products in the form of SCFAs, and intestinal inflammation. We showed a significant correlation between nutritional status, as measured by BMI and albumin levels, with well-known inflammatory parameters such as complete blood count, CRP, and proinflammatory cytokines. Knowing the exact bacterial composition in patients with UC and CD seems crucial for better use of SCFA supplementation and nutritional therapy in the treatment of these conditions. Our study shows that there is a need for the better use of nutritional assessment in the daily care of patients with IBD. Moreover, it is necessary to clarify nutritional recommendations specifically for patients with CD and UC.

## Figures and Tables

**Figure 1 nutrients-14-03960-f001:**
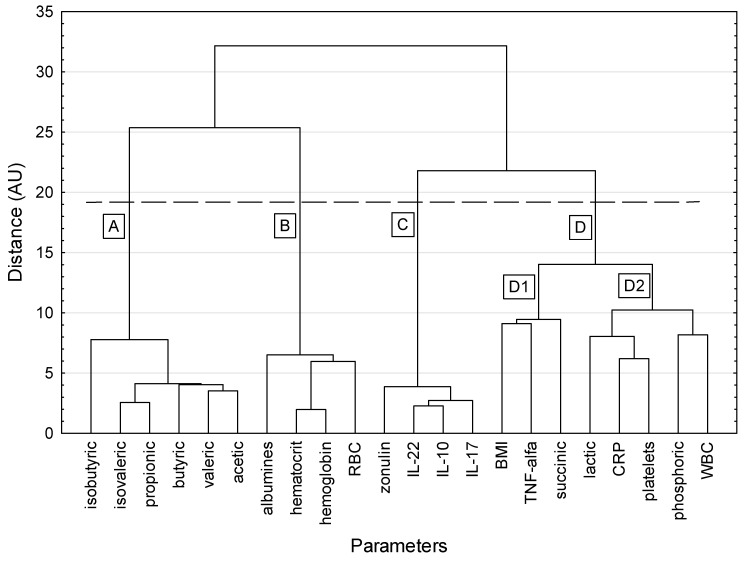
A dendrogram of similarity between studied parameters (method of grouping: Ward agglomeration; function of the distance: Euclidean distance; the dashed horizontal line indicates that grouping was stopped according to the Mojena’s rate). BMI: body mass index; CRP: C-reactive protein; RBC: red blood cells; WBC: white blood cells.

**Figure 2 nutrients-14-03960-f002:**
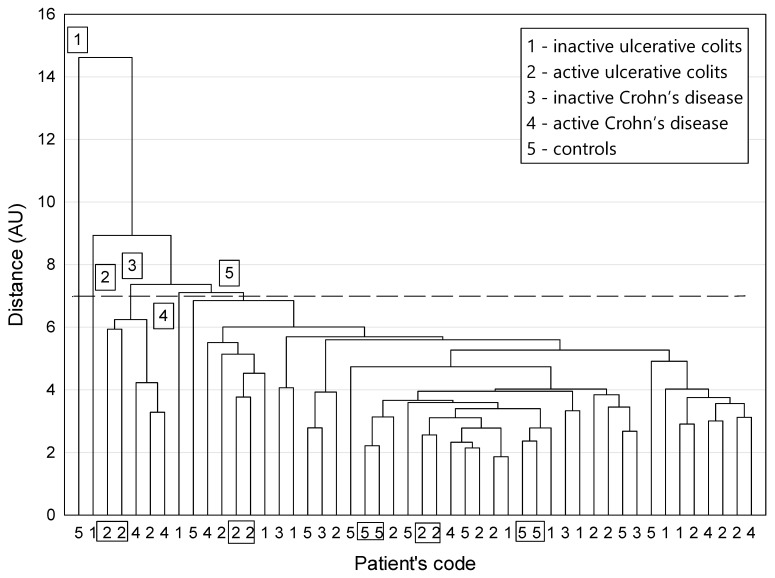
A dendrogram of similarity among study participants (method of grouping: average linkage procedure; function of the distance: Euclidean distance; the dashed horizontal line indicates that grouping was stopped according to the Mojena’s rate); patients were labeled according to the classification code; the determined clusters of patients were labeled with numbers: 1–5; rectangles encompass patients belonging to two-element clusters.

**Table 1 nutrients-14-03960-t001:** Characteristics of patients with ulcerative colitis (UC), Crohn’s disease (CD), and controls.

Characteristics	UC	CD	Controls	*p* Value *
number of participants, *n* (%)	26 (54)	10 (20)	12 (26)	-
sex, *n* (male/female)	17/9	5/5	9/3	-
age, years	35 (18–71)	28 (18–43)	23.5 (22–77)	0.078
active disease, *n* (%)	18 (37.5)	6 (12.5)	-	0.599 **
BMI, kg/m^2^	21.5 (15.7–29.0)	20.2 (13.8–27.8)	21.9 (19.1–30.7)	0.322
hemoglobin, g/dL	13.3 (7.3–15.9)	11.6 ^a^ (7.2–14.4)	14.7 ^a^ (10.2–16.9)	<0.05
albumin, g/L	41.1 (26.4–50.3)	37.1 ^a^ (24.0–41.0)	41.0 ^a^ (32.0–48.9)	<0.05
CRP, mg/L	6.7 (1.0–154.0)	15.8 (1.0–145.0)	1.0 (1.0–6.0)	0.012
calprotectin, µg/g	1261 ^a^ (1–2223)	861 ^b^ (1–2179)	15 ^a,b^ (0–15)	0.946 ***

Data are presented as number (percentage) of patients or as median (min-max). CRP = C-reactive protein; BMI = body mass index. * *p* value determined by the Kruskal-Wallis test when comparing all three groups; if the Kruskal-Wallis test showed a significant difference, the *p* value calculated by the Dunn test was given, and the upper-index letter “a” or “b” indicates the differing groups. ** *p* value determined by the Pearson chi-squared test *** *p* value determined by the Mann-Whitney test, which was applied only to UC and CD groups, as there were only two observations in the control group.

**Table 2 nutrients-14-03960-t002:** Concentrations of organic acids in patients with ulcerative colitis (UC), Crohn’s disease (CD), and controls.

Organic Acid	UC (*n* = 26)	CD (*n* = 10)	Controls (*n* = 12)	*p* Value *
succinic, µg/g	297.2 (149.4; 699.3)	614.1 (170.4; 711.6)	425.8 (103.5; 949.5)	0.764
acetic, µg/g	1028.4 ^a^ (731.8; 1292.8)	1013.3 (406.1; 1454.4)	2001.2 ^a^ (931.7; 2484.9)	<0.05
lactic, µg/g	825.3 (289.2; 1591.0)	668.6 (119.8; 1622.4)	242.0 (32.0; 1559.8)	0.514
propionic, µg/g	285.8 (179.4; 704.1)	347.8 (214.4; 693.3)	522.5 (360.7; 808.1)	0.259
butyric, µg/g	236.3 (42.2; 534.7)	49.3 ^a^ (<LOD; 220.9)	449.5 ^a^ (172.5; 703.7)	<0.01
isobutyric, µg/g	33.6 (<LOD; 73.1)	46.5 (<LOD; 82.4)	47.5 (33.3; 86.6)	0.337
valeric, µg/g	<LOD ^a^ (<LOD; <LOD)	<LOD (<LOD; 40.7)	29.1 ^a^ (13.3; 57.9)	<0.05
isovaleric, µg/g	50.6 (<LOD; 112.6)	47.8 (<LOD; 73.1)	62.7 (32.0; 116.5)	0.675
phosphoric, µg/g	818.7 (5.52; 2139.67)	882.4 (331.5; 1706.6)	583.58 (5.52; 1181.99)	0.180

Data are presented as medians with lower and upper quartiles. LOD = lower limit of detection * *p* value determined by the Kruskal-Wallis test when comparing all three groups; if the Kruskal-Wallis test showed a significant difference, the *p* value calculated by the Dunn test was given, and the upper-index letter “a” indicates the differing groups.

**Table 3 nutrients-14-03960-t003:** Cytokine and zonulin levels in patients with ulcerative colitis (UC), Crohn’s disease (CD), and controls.

Serum Levels	UC (*n* = 26)	CD (*n* = 10)	Controls (*n* = 16)	*p* Value *
TNF-α, µg/g	6.1 (1.64; 8.83)	6.3 ^a^ (5.83; 10.83)	2.0 ^a^ (0.35; 4.64)	<0.01
IL-17, µg/g	116.1 (56.21; 119.54)	144.9 (66.4; 264.36)	52.9 (36.2; 108.95)	0.147
IL-10, µg/g	30.5 (16.99; 40.77)	35.5 (15.22; 59.5)	17.4 (12.84; 32)	0.230
IL-22, µg/g	36.3 (15.37–52.43)	62.7 (23.93; 100.97)	21.8 (16.17; 44.03)	0.256
IL-6, µg/g	<LOD (<LOD; 0.44)	<LOD (<LOD; 7.49)	<LOD (<LOD; <LOD)	0.126
zonulin, µg/g	6.0 (1.62; 42.71)	15.5 (0.97; 41.19)	2.8 (1.34; 8.04)	0.716

Data are presented as medians with lower and upper quartiles. LOD = lower limit of detection; TNF-α = tumor necrosis factor α. * *p* value determined by the Kruskal-Wallis test when comparing all three groups; if the Kruskal-Wallis test showed a significant difference, the *p* value calculated by the Dunn test was given, and the upper-index letter “a” indicates the differing groups.

## Data Availability

The data presented in this study are available on request from the corresponding author.

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
