# Peer review of "Application of the Clustering Technique to Multiple Nutritional Factors Related to Inflammation and Disease Progression in Patients with Inflammatory Bowel Disease"

_nutrients, 2022, doi:10.3390/nu14193960_

Round 1
Reviewer 1 Report
Paper written well describing diet in IBD
please describe in more detail about cluster analysis; some people do not understand what it means
I am not sure how relevant this paper is to clinical work. It does not bring any new ideas to diet and IBD.
Author Response
Response to Review Comments
We would like to thank the editors and reviewers for their comprehensive review. We have revised our paper in line with the suggestions and comments. Our responses are included below.
Reviewers' comments:
Reviewer 1
- Please describe in more detail about cluster analysis; some people do not understand what it means.
Answer:
We would like to thank for this suggestion. We have described cluster analysis in more detail to make it more comprehensible to readers who are not familiar with this statical method. Proper changes have been made in the text in section 2.5.2.
- I am not sure how relevant this paper is to clinical work. It does not bring any new ideas to diet and IBD.
Answer:
Thank you for your comment. The results of our study demonstrated associations between several biochemical parameters of inflammation and nutritional parameters that are used in daily clinical practice. Despite growing knowledge about the role of diet and nutritional status in the pathogenesis and course of IBD, this aspect is still often overlooked in daily practice [1]. Our study, conducted on IBD patients with colon involvement, shows concrete examples of the relationship between routinely used inflammatory parameters such us serum leukocytes, platelets, C-reactive protein, as well as simple nutritional assessment such us body mass index (BMI), body fluctuation and albumin levels. We would like to emphasize that we evaluated all markers of disease activity that should be used in the routine monitoring of IBD patients according to European Crohn’s and Colitis Organisation guidelines [2]. Moreover, observed associations allow for a better interpretation of the laboratory results obtained in the context of a patient's nutritional status. Finally, they highlight the need to include nutritional evaluation in the routine monitoring of IBD patients.
The demonstrated relationships between inflammatory markers and nutritional parameters also becomes significant in the light of reports that not only underweight but also obesity seems to be an important risk factor for IBD disease severity and clinical outcomes because adipose tissue, especially visceral, significantly contributing to inflammatory processes [3].
Further research on these relationships is needed, enabling the selection of the best markers to assess the risk and the effectiveness of the treatment in this group of patients.
Reference:
[1] Aleksandrova K., Romero-Mosquera B., Hernandez V. Diet, gut microbiome and epigenetics: Emerging links with inflammatory bowel diseases and prospects for management and prevention. Nutrients. 2017; 9:962. doi: 10.3390/nu9090962.
[2] Maaser, C.; Sturm, A.; Vavricka, S.R.; Kucharzik, T.; Fiorino, G.; Annese, V.; Calabrese, E.; Baumgart, D.C.; Bettenworth, D.; Borralho Nunes, P.; Burisch, J.; Castiglione, F.; Eliakim, R.; Ellul, P.; González-Lama, Y.; Gordon, H.; Halligan, S.; Katsanos, K.; Kopylov, U.; Kotze, P.G.; Krustinš, E.; Laghi, A.; Limdi, J.K.; Rieder, F.; Rimola, J.; Taylor, S.A.; Tolan, D.; van Rheenen, P.; Verstockt, B.; Stoker, J.; European Crohn’s and Colitis Organisation [ECCO] and the European Society of Gastrointestinal and Abdominal Radiology [ESGAR]. ECCO-ESGAR Guideline for Diagnostic Assessment in IBD Part 1: Initial diagnosis, monitoring of known IBD, detection of complications. J Crohn’s Colitis 2019, 13, 144–164. doi: 10.1093/ecco-jcc/jjy113
[3] Karaskova E, Velganova-Veghova M, Geryk M, Foltenova H, Kucerova V, Karasek D. Role of Adipose Tissue in Inflammatory Bowel Disease. Int J Mol Sci. 2021 Apr 19;22(8):4226.
Reviewer 2 Report
Using a cluster analysis, the authors investigated the correlations between nutritional status, short-chain fatty acid (SCFA) levels and proinflammatory and anti-inflammatory cytokines in patients with inflammatory bowel disease (IBD).
Minor revisions need to be done.
1) General Revision:
- Typography: the authors should read thoroughly their manuscript and check: 1) space between words; 2) English of some sentences.
2) Introduction section:
- We suggest explaining better how cytokines are modulated by SCFAs. In this context, add references also regarding Interleukin-22 (IL-22) and tumor necrosis factor-α (TNF- α ) (line 51).
3) Materials and Methods section:
- Please, specify the sex of patients under study.
- Please, briefly explain the methodology used for the identification of Fecal Organic Acids.
- Please, briefly explain Ward agglomeration method.
Author Response
Response to Review Comments
We would like to thank the editors and reviewers for their comprehensive review. We have revised our paper in line with the suggestions and comments. Our responses are included below.
Reviewer 2
- General Revision: Typography: the authors should read thoroughly their manuscript and check: 1) space between words; 2) English of some sentences.
Answer:
Thank you for your comment. The manuscript has been edited by a professional English language editor and necessary revisions have been applied.
- Introduction section: We suggest explaining better how cytokines are modulated by SCFAs. In this context, add references also regarding Interleukin-22 (IL-22) and tumor necrosis factor-α (TNF- α) (line 51).
Answer:
Thank you for this suggestion. We have added a few sentences explaining how SCFAs modify the production of these cytokines in the Introduction Section (line 52-57).
- Materials and Methods section:
- Please, specify the sex of patients under study.
- Please, briefly explain the methodology used for the identification of Fecal Organic Acids.
- Please, briefly explain Ward agglomeration method.
Answer:
We would like to thank for this suggestion. The gender specificity of the study participants has been added to the manuscript (Table 1). As suggested, we have modified a paragraph describing methodology of SCFAs determination in the Materials and Methods Section. Proper changes about ward agglomeration method have been made in the text in the Materials and Methods Section.
Round 2
Reviewer 1 Report
Good review